# TOWARDS OUT-OF-FEDERATION GENERALIZATION IN FEDERATED LEARNING

## ABSTRACT

Federated Learning (FL) is widely employed to tackle distributed sensitive data. Existing methods primarily focus on addressing in-federation data heterogeneity. However, we observe that they can suffer from significant performance degradation when applied to unseen clients for out-of-federation (OOF) generalization. The recent attempts to address generalization to unseen clients generally fail to scale up to large-scale distributed settings due to high communication overhead and convergence difficulty. And the communication efficient methods often yield poor OOF robustness. To achieve OOF-resiliency in a scalable manner, we propose Topology-aware Federated Learning (TFL) that leverages client topology - a graph representing client relationships - to effectively train robust models against OOF data. We formulate a novel optimization problem for TFL, consisting of two key modules: Client Topology Learning, which infers the client relationships in a privacy-preserving manner, and Learning on Client Topology, which leverages the learned topology to identify influential clients and harness this information into the FL optimization process to efficiently build robust models. Empirical evaluation on a variety of real-world datasets verifies TFL's superior OOF robustness and communication efficiency. Our source code is available at https://anonymous.4open.science/r/TFL-8390.

## 1 INTRODUCTION

Modern industries, ranging from healthcare to finance, accumulate vast amounts of sensitive information, including personal records and proprietary data, which are often distributed across different institutions and subject to strict privacy regulations (CCPA, 2018). This fragmentation of data presents a significant challenge in centralizing it to develop powerful ML models. Federated Learning (FL) has emerged as a promising solution to tackle this issue, enabling multi-institutional collaboration by distributing model training to data owners and aggregating results on a centralized server (McMahan et al., 2017). This data-decentralized approach harnesses the collective intelligence of all participating nodes to build a model that is potentially more robust and generalizable. Existing robust FL methods (Li et al., 2020b; Deng et al., 2020) primarily focus on learning a global model with good average or worst-case performance, addressing *in-federation (IF)* data het-

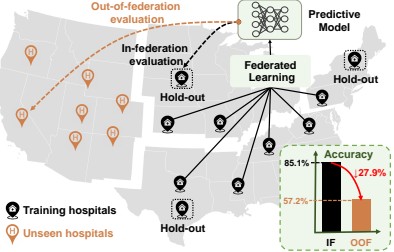

Figure 1: Federated learning of predictive model for patient mortality prediction from distributed healthcare datasets (Pollard et al., 2018). A model that is highly accurate on in-federation (IF) data can fail catastrophically when presented with out-of-federation (OOF) data.

erogeneity. However, these methods can fail catastrophically on *out-of-federation (OOF)* clients, *i.e.,* clients outside the collaborative federation. The OOF clients pose significant generalization challenges, as FL models may encounter *unseen* distributions outside their training space (Pati et al., 2022a). Our empirical study shows that existing methods suffer from significant performance degradation when applied to unseen clients for OOF generalization (see an example in Figure 1).

There have been recent attempts to address the challenge of unseen clients through client augmentation (Liu et al., 2021) and client alignment (Nguyen et al., 2022). However, as shown in Figure 2, our empirical evaluation shows that the existing method suffers from a tradeoff between

OOF-resiliency and communication efficiency. Client augmentation-based methods often necessitate extensive client-server communication, which is not scalable for large-scale settings (Zhou et al., 2023). Client alignment-based methods align the latent representation to a reference distribution. Though this strategy is scalable, it has been observed with limited OOF robustness (Bai et al., 2023). A question naturally arises: How to obtain good OOF robustness in a scalable manner?

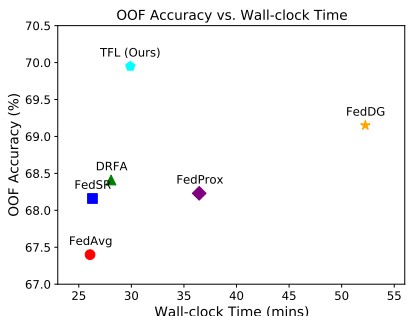

Figure 2: OOF accuracy vs. wall-clock time on PACS dataset (Li et al., 2017). We see a clear tradeoff between OOF robustness and scalability (communication/computation).

We aim to trigger OOF-resiliency while being communication efficient by leveraging client topology. Client topology, a graph representing client relationships, allows for using graph mining techniques (Saxena & Iyengar, 2020; Lü & Zhou, 2011; Nascimento & De Carvalho, 2011) to derive insights into client data distributions. It can be used to identify "influential" clients that are representative of the training clients, containing distributions more likely to be encountered in OOF clients. For instance, an influential client could be a regional hospital that aggregates a diverse range of patient data. This hospital's data encapsulates a rich repository of information, mirroring the variety and complexity of data that could be seen in OOF scenarios. Leveraging these influential clients as priority contributors in the training rounds can facilitate the model in learning from the most representative data, thereby potentially enhancing its OOF robustness. On the other hand, by reducing unnecessary communication with non-influential clients, communication costs can be significantly reduced.

With this client topology-based design rationale, we formulate a novel optimization problem for OOF generalization, which simultaneously optimizes the robust models and the client topology. To solve this optimization problem, we propose Topology-aware Federated Learning (TFL), which consists of two key steps. 1) **Client Topology Learning**: Inferring the client topology with respect to data privacy. 2) **Learning on Client Topology**: Leveraging the learned topology to build a robust model. The first step learn a client topology by promoting a correlation with model similarity. In the second step, a robust model is efficiently optimized by harnessing the client's influential information to regularize a distributed robust optimization process.

Our main contributions are as follows: Firstly, we introduce the *Topology-aware Federated Learning* (TFL) framework, a principled approach designed to enhance FL's out-of-federation (OOF) robustness. TFL utilizes client relationships to develop robust models against OOF data. Secondly, we design an iterative *client topology leaning* and *learning on client topology* approach to solve TFL. Finally, we have curated two OOF benchmarks using real-world healthcare data, offering valuable testbeds for subsequent research. Through extensive experiments on these and standard benchmarks, we verify TFL's superior OOF-resiliency and scalability.

## 2 PRELIMINARIES

**Federated learning (Average-case formulation).** The standard FL involves collaboratively training a global model leveraging data distributed at $K$ clients. Each client $k$ ($1 \leq k \leq K$) has its own data distribution $\mathcal{D}_k(x, y)$, where $x \in \mathcal{X}$ is the input and $y \in \mathcal{Y}$ is the label, and a dataset with $n_k$ data points: $\widehat{\mathcal{D}}_k = \{(x_k^n, y_k^n)\}_{n=1}^{n_k}$. Local data distributions $\{\mathcal{D}_k\}_{k=1}^{K}$ could be the same or different across the clients. FL aims to learn a global model $\theta$ by minimizing the following objective function:

$$\min_{\theta \in \Theta} F(\theta), \text{ where } F(\theta) := \sum_{k=1}^{K} p_k f_k(\theta), \tag{1}$$

where $f_k(\theta)$ is the local objective function of client $k$. The local objective function is often defined as the empirical risk over local data, *i.e.,* $f_k(\theta) = \mathbb{E}_{(x,y) \sim \widehat{\mathcal{D}}_k}[\ell(\theta; x, y)] = \frac{1}{n_k} \sum_{n=1}^{n_k} \ell(\theta; x_k^n, y_k^n)$. The term $p_k$ ($p_k \geq 0$ and $\sum_k p_k = 1$) specifies the relative importance of each client, with two settings being $p_k = \frac{1}{N}$ or $p_k = \frac{n_k}{N}$, where $N = \sum_k n_k$ is the total number of samples.

**Distributionally robust federated learning (Worst-case formulation).** While Equation 1 can build a global model with good *average performance* on in-federation clients, it may not neces-

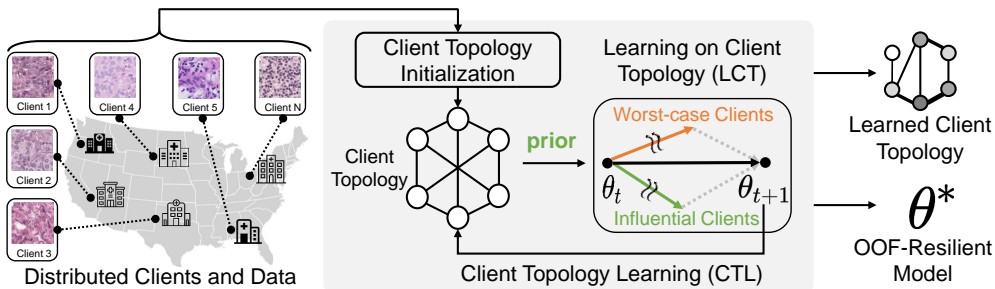

Figure 3: Overview of Topology-aware Federated Learning (TFL). TFL contains two alternating steps: *client topology learning* (CTL) and *learning on client topology* (LCT). **CTL** learn the client topology that describes the relationships between local clients. We leverage model weights to construct a graph by measuring model similarity. **LCT** leverage the learned client topology to achieve better OOF robustness. We identify the influential client and then use the influential client as prior knowledge to regularize a distributionally robust optimization framework. In this way, the optimization process can balance the "worst-case" client and the "influential" client to avoid overly pessimistic models with compromised OOF-resiliency.

sarily guarantee good performance in the presence of heterogeneous data. In the real world, data could be statistically heterogeneous due to different data acquisition protocols or various local demographics (Rieke et al., 2020). Thus, the local data distributions may deviate significantly from the average distribution, implying that an "average" global model can fail catastrophically under distributional drift. To tackle this issue, distributionally robust optimization (Staib & Jegelka, 2019) has been adapted to FL, resulting in distributionally robust federated learning (Deng et al., 2020; Reisizadeh et al., 2020). The formulation of this new problem is as follows:

$$\min_{\theta \in \Theta} \max_{\boldsymbol{\lambda} \in \Delta_K} F(\theta, \boldsymbol{\lambda}) := \sum_{k=1}^{K} \lambda_k f_k(\theta), \tag{2}$$

where $\boldsymbol{\lambda}$ is the global weight for each local loss function and $\Delta_K$ denotes the $K-1$ probability simplex. Intuitively, Equation 2 minimizes the maximal risk over the combination of empirical local distributions, and therefore the worst-case clients would be prioritized during training.

While this framework has the potential to address distribution shifts (Mohri et al., 2019; Deng et al., 2020), directly implementing it for OOF resiliency may yield suboptimal models. This approach heavily relies on *worst-case clients*, those with large empirical risks, to develop robust models. However, these clients may not necessarily be the *influential ones* that are representative of clients. In some cases, it is possible that this approach overly focuses on "outlier" clients, clients that are significantly different from most of the training clients, leading to models with limited OOF robustness. Therefore, we argue that, to build optimal OOF-resilient models, the optimization process should focus on not only the worst-case but also the influential clients.

## 3 METHODOLOGY

In this section, we introduce our *Topology-aware Federated Learning* (TFL) framework. TFL aims to leverage client topology to boost the model's OOF robustness. We model client topology using an undirected graph (Vanhaesebrouck et al., 2017). In the graph, nodes correspond to clients, and edges reflect clients' connectivity. Let $\mathcal{G} = (V, E, W)$ denote the client topology, where $V$ is the node set with $|V| = K$, $E \subseteq V \times V$ is the edge set and $W \in \mathbb{R}^{K \times K}$ is the adjacency matrix. An edge between two clients $v_k$ and $v_l$ is represented by $e_{k,l}$ and is associated with a weight $w_{k,l}$.

**Optimization problem.** As the client topology is often not readily available, we propose to jointly optimize the client topology and the robust model by solving the following problem:

$$\min_{\substack{\theta \in \Theta \\ w \in W}} \max_{\boldsymbol{\lambda} \in \Delta_K} F(\theta, \boldsymbol{\lambda}, W) := \sum_{k=1}^{K} \lambda_k f_k(\theta) - \frac{\gamma}{2} \sum_{k \neq l} w_{k,l} \, \text{sim}(v_k, v_l),$$

$$\text{s.t. } \mathcal{D}(\boldsymbol{\lambda} \parallel \boldsymbol{p}) \leq \tau, \text{ where } \boldsymbol{p} = \phi(W), \tag{3}$$

where $\gamma$ is a trade-off hyperparameter. $\phi$ denote graph measures, *e.g,* centrality measure (Saxena & Iyengar, 2020). The function "sim" indicates any chosen similarity function, including but not limited to cosine similarity, dot product, $\ell_2$, or $\ell_1$. In the objective function, the first term follows the same spirit of Equation 2 to adopt a minimax robust optimization framework. The difference is that it minimizes the risk over not only the worst-case but also the influential clients. The second term learn a client topology by measuring the pair-wise client similarity.

Our formulation stands apart from existing work in two respects. Firstly, it employs client topology to explicitly model the client relationships. Analyzing this topology facilitates the identification of influential clients that are crucial for developing strong OOF generalization. Secondly, our formulation enables seamless integration of client topology into the optimization process, guaranteeing that the model assimilates insights from the most significant clients.

In Equation 3, simultaneously updating both the client topology $W$ and model parameters $\theta$ is infeasible as local clients do not have access to other clients' data. Thus, we propose to solve this problem using an alternating two-step approach: **Client Topology Learning** (updating $W$) and **Learning on Client Topology** (updating $\lambda$ and $\theta$) (see Figure 3). The following sections will provide further details on these two steps.

### 3.1 CLIENT TOPOLOGY LEARNING

Our goal is to learn the client topology that accurately describes the characteristics of local data, thereby capturing the underlying client relationships. Conventional approaches typically adopt similarity-based (Chen et al., 2020a) or diffusion-based (Zhu et al., 2021; Mateos et al., 2019) methods to estimate the graph structure from data. However, most methods require centralizing training data on a single machine, raising privacy concerns. Therefore, the key question is how to learn the client topology while respecting data privacy.

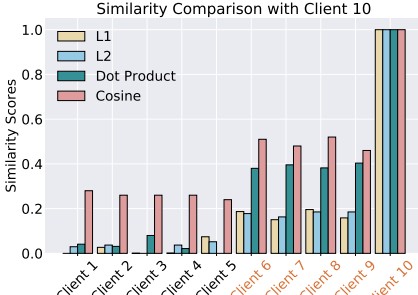

We propose to utilize model weights to learn client topology. Intuitively, when two clients have similar data distributions, their corresponding models should be more similar. It is possible to obtain data distribution relationships with model similarity. While the feasibility of this idea is supported by literature (Yu et al., 2022), we intend to empirically verify whether it holds in our setting. We use four types of similarity measures, including $\ell_1$-based, $\ell_2$-based, dot product-based, and cosine similarity-based metrics. We conduct experiments on PACS where clients 6 to 10 share similar data distributions. Then, we measure the similarity between client 10 and all other clients. For a clear comparison, we normalized all scores into $[0, 1]$. The results are shown in Figure 4. We observe that the similarity scores of clients with the same data distributions are much higher than other clients. And these distinguishable similarity scores are consistent across all metrics.

Figure 4: Model similarity under different metrics. Client models are trained using the same algorithm and hyperparameters. Clients 6 to 10 share similar data distributions. We observe that *clients with similar data distribution tend to have more similar models.*

In summary, utilizing model weights to learn client topology offers two merits. First, models can be freely shared among clients, addressing privacy issues of client topology learning. Second, since models are trained on local data, their similarity measures the similarity between local data distributions. We formulate client topology learning as follows:

$$\min_{w \in W} -\sum_{k \neq l} w_{k,l} \operatorname{sim}(\theta_k, \theta_l) + \|W\|_0 \,, \tag{4}$$

where $\theta$ denotes the model of local clients. This objective ensures that similar clients have large edge weights. Our formulation is aligned with the common network homophily assumption that *edges tend to connect similar nodes* (Newman, 2018). To avoid learning a fully connected graph with trial edges, we enforced graph sparsity by penalizing the $l_0$-norm of the adjacency matrix $\|W\|_0$. We implement the sparsity term using an implicit method, *i.e.,* hard thresholding on $W$ to construct $\epsilon$-graph (Zhu et al., 2021). Specifically, we mask out (*i.e.,* set to zero) those elements in $W$ smaller than a non-negative threshold $\epsilon$.

## 3.2 LEARNING ON CLIENT TOPOLOGY

The learned client topology captures the relation among local clients. We aim to leverage such relations to develop TFL for better OOF-resiliency. Recall that, to tackle distribution shift, distributionally robust federated learning (DRFL) assumes that the target distribution lies within an arbitrary mixture of training distributions: $\sum_{k=1}^{K} \lambda_k \mathcal{D}_k$. DRFL builds OOF-resilient models by minimizing the *worst-case* risk over an uncertainty set of possible target distribution $Q := \{\sum_{k=1}^{K} \lambda_k \mathcal{D}_k \mid \boldsymbol{\lambda} \in \Delta_K\}$. However, DRFL mainly emphasizes the worst-case distribution while ignoring the *influential* ones that are representative of training clients, yielding overly pessimistic models with compromised OOF resiliency (Hu et al., 2018; Frogner et al., 2021).

We leverage client topology to construct an uncertainty set which can better approximate the unseen distribution. Our insight is to optimize the model for both the worst-case and influential distributions. The key challenge is how to identify the influential distribution. Our idea is to use graph centrality as the criterion to choose influential distributions. *Graph centrality* is widely used in social network analysis (Newman, 2005) to identify the influential person by measuring how much information propagates through each entity. We introduce *client centrality* to identify influential clients, which can be calculated by graph measurements such as degree, closeness, and betweenness. Specifically, we first calculate the centrality of each client in $\mathcal{G}$ as the *topological prior* $\boldsymbol{p}$. Then we use $\boldsymbol{p}$ to constraint the uncertainty set $Q$ by solving the following minimax optimization problem:

$$\min_{\theta \in \Theta} \max_{\boldsymbol{\lambda} \in \Delta_K} F(\theta, \boldsymbol{\lambda}) := \sum_{k=1}^{K} \lambda_k f_k(\theta), \ \ \text{s.t.} \ \ \mathcal{D}(\boldsymbol{\lambda} \parallel \boldsymbol{p}) \leq \tau. \tag{5}$$

The topological constraint enforces the optimization process to focus on not only the worst-case but also influential clients. Here $\mathcal{D}$ is an arbitrary distributional distance measure (In this paper, we choose $\ell_2$ distance). It is worth noting that both federated averaging (FedAvg (McMahan et al., 2017)) (Equation 1) and DRFL (Equation 2) are special cases of TFL: When $\tau = 0$ and the prior $\boldsymbol{p}$ is a uniform distribution, Equation 5 minimizes the average risk over local clients, which is identical to FedAvg; When $\tau \to \infty$, Equation 5 only prioritizes the worst-case clients, degrading to DRFL.

The above optimization problem is typically nonconvex, and methods such as SGD cannot guarantee constraint satisfaction (Robey et al., 2021). To tackle this issue, we leverage the Lagrange multiplier and KKT conditions (Boyd et al., 2004) to convert it into unconstrained optimization:

$$\min_{\theta \in \Theta} \max_{\boldsymbol{\lambda} \in \Delta_K} F(\theta, \boldsymbol{\lambda}, \boldsymbol{p}) := \sum_{k=1}^{K} \lambda_k f_k(\theta) - q\mathcal{D}(\boldsymbol{\lambda} \parallel \boldsymbol{p}), \tag{6}$$

where $q$ is the dual variable. Then we solve the primal-dual problem by alternating between gradient descent and ascent:

$$\theta^{t+1} = \theta^t - \eta_\theta^t \nabla_\theta F(\theta, \boldsymbol{\lambda}), \ \ \boldsymbol{\lambda}^{t+1} = \mathcal{P}_{\Delta_K}(\boldsymbol{\lambda}^t + \eta_\lambda^t \nabla_\lambda F(\theta, \boldsymbol{\lambda})), \tag{7}$$

where $\eta^t$ is the step size. $\mathcal{P}_{\Delta_K}(\boldsymbol{\lambda})$ projects $\boldsymbol{\lambda}$ onto the simplex $\Delta_K$ for regularization.

**Influential client.** We identify the influential clients by calculating betweenness centrality. Betweenness centrality measures how often a node is on the shortest path between two other nodes in the graph $\mathcal{G}$. It has been revealed that the high betweenness centrality nodes have more control over the graph as more information will pass through them (Freeman, 1977). The betweenness centrality of client $k$ is given by the expression of $c_k = \sum_{s \neq k \neq t \in [K]} \frac{\sigma_{st}(k)}{\sigma_{st}}$, where $\sigma_{st}$ is the total number of shortest path from node $s$ to node $t$ ($(s, t)$-paths) and $\sigma_{st}(k)$ is the number of $(s, t)$-paths that pass through node $k$. Then we apply softmax to normalize client centrality $c_k$ to obtain client topological prior $p_k = \exp(c_k) / \sum_{k=1}^{K} \exp(c_k)$.

**Discussion. Handling high-dimensional models.** Large models are getting more attention in FL (Zhuang et al., 2023). Its high dimensionality might raise computational concerns when calculating model similarity. For better computation efficiency, we can leverage only partial model parameters, *e.g.*, the last few layers. We empirically show that using partial parameters basically does not affect the OOF performance. **Client topology learning in large-scale settings**. Client topology learning requiris $\mathcal{O}(N^2)$ computation complexity for $N$ clients. This quadratic complexity

is prohibitively expensive in cross-device FL, where thousands or millions of clients are involved. We argue that computation costs can be significantly reduced via client clustering (Sattler et al., 2020). By partitioning the clients into clusters, the total number of "clients" is reduced, allowing for cluster-level client topology learning with reduced computation costs. We empirically show that client clustering can significantly reduce computation costs (**see results in Supplementary F**).

## 4 EXPERIMENTS

### 4.1 DATASETS AND BASELINES

**Datasets.** Our method is evaluated on real-world datasets (①eICU, ②FeTS, ③TPT-48) and standard benchmarks (④CIFAR-10/-100, ⑤PACS), spanning a wide range of tasks including classification, regression, and segmentation. Evaluations encompass both out-of-federation (datasets ①-③, ⑤) and in-federation (datasets ③-④) scenarios. Further dataset details are available in Supplementary B.

① **eICU** (eICU Collaborative Research Database) (Pollard et al., 2018) is a large-scale multi-center critical care database. It contains high granularity critical care data for over $200,000$ patients admitted to 208 hospitals across the United States. Each hospital is considered an individual client. he generalization task could be deploying models trained on hospitals from the SOUTH region to those in the WEST. The evaluation metric for patient mortality prediction is ROC-AUC.

② **FeTS** (Federated Tumor Segmentation Dataset) (Pati et al., 2022b) is a multi-institutional medical imaging dataset. It comprises clinically acquired multi-institutional MRI scans of glioma. A subset of the original data was used, comprising 358 subjects from 21 distinct global institutions. The associated task is to identify and delineate brain tumor boundaries. Each institution is a client. The evaluation metric is Dice Similarity Coefficient (DSC $\uparrow$).

③ **TPT-48** (Vose et al., 2014) contains the monthly average temperature for the 48 contiguous states in the US from 2008 to 2019. The task is to predict the next six months' temperature given the first six months' temperature. For TPT-48, we consider two generalization tasks: (1) E(24) $\rightarrow$ W(24): Using the 24 eastern states as IF clients and the 24 western states as OOF clients; (2) N(24) $\rightarrow$ S(24): Using the 24 northern states as IF clients and the 24 southern states as OOF clients. The evaluation metric is Mean Squared Error (MSE $\downarrow$).

④ **CIFAR-10/-100** (Krizhevsky & Hinton, 2009) are the most commonly used benchmarks in FL literature. In these datasets, we use Dirichlet distribution (Hsu et al., 2019) to partition the dataset into the heterogeneous setting with 25 and 50 clients.

⑤ **PACS** (Li et al., 2017) contains $9,991$ images from four domains: art painting, cartoon, photo, and sketch. The task is seven-class classification. For PACS, we evenly split each domain into 5 subsets, yielding 20 subsets, and we treat each subset as a client. We followed the common "leave-one-domain-out" experiment, where 3 domains are used (15 clients) for training and 1 domain (5 clients) for testing. Model performance is evaluated by classification accuracy.

**Baselines**. We compare with ① FedAvg (McMahan et al., 2017) and ② FedProx (Li et al., 2020b). These two baselines are the most referenced methods. ③ DRFA (Deng et al., 2020) is the latest work that adopts the federated distributionally robust optimization framework. ④ FedSR (Nguyen et al., 2022) is the most recent work that tackles FL's generalization to unseen clients. We did not compare with FedDG (Liu et al., 2021) as it requires the sharing of data in the frequency space with each other. This can be viewed as a form of data leakage (Bai et al., 2023; Nguyen et al., 2022). We provide implementation details on model architecture and hyperparameters in Supplementary E.

### 4.2 EVALUATION ON OOF-RESILIENCY

**Takeaway 1: Learning on client topology improves OOF robustness**. We evaluate our method on different datasets and summarize the results in Tables 1, 2, and 3. We make the following observation: our method improves the model's OOF-resiliency on both standard and real-world datasets. From Table 1, our method performs $2.25\%$ better than the federated robust optimization method (DRFA) and $2.63\%$ better than the federated domain generalization method (FedSR). From Table 2, our method performs $2.11\%$ better than DRFA and $2.03\%$ better than FedSR. Lastly, from Table 3, our method performs $1.98\%$ better than DRFA and $3.00\%$ better than FedSR. Overall, our method shows consistently superior OOF robustness than state-of-the-art across all the evaluated datasets. We also visualize the segmentation results of FeTS in Figure 6. We observe that

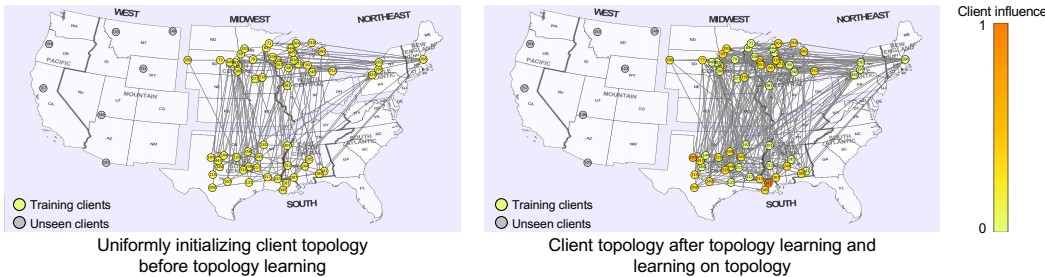

Figure 5: Visualization of client topology on the eICU dataset. Hospitals from MIDWEST, SOUTH, and NORTHEAST collaboratively train a global model, which is subsequently evaluated on hospitals in the WEST. We observe that the learned client topology becomes denser, revealing underlying relationships that were previously unknown. Furthermore, the identified "influential" hospitals are more from the MIDWEST and SOUTH rather than NORTHEAST. This observation is rational, given the geographical proximity of these two regions to the target evaluation region, the WEST.

our method delivers a higher quality of segmentation, suggesting increased reliability for real-world healthcare applications that have to contend with diverse local demographics.

**Visualization of client topology**. We also visualize the learned client topology of the eICU dataset in Figure 5. We observe that the learned client topology helps to identify the "influential" client. The figure shows that the important clients predominantly originate from the MIDWEST and SOUTH, while none from the NORTHEAST are influential.

Table 1: Accuracy on the PACS dataset. We conduct experiments using a *leave-one-domain-out* approach, meaning each domain serves as the evaluation domain in turn. Existing methods typically consider each domain as an individual client (Liu et al., 2021; Nguyen et al., 2022). To simulate a large-scale distributed setting, we further divide each domain into 5 subsets and treat each subset as a separate client. The total number of clients is 20. The reported numbers are from three independent runs. *Our method outperformed others across all experimental settings*.

| | Models | Backbone | PACS | | | | |
|---|---|---|---|---|---|---|---|
| | | | A | C | P | S | Average |
| Centralized | DGER (Zhao et al., 2020) | ResNet18 | 80.70 | 76.40 | 96.65 | 71.77 | 81.38 |
| Methods | DIRT-GAN (Nguyen et al., 2021) | ResNet18 | 82.56 | 76.37 | 95.65 | 79.89 | 83.62 |
| | FedAvg | ResNet18 | 55.83±0.31 | 61.37±0.66 | 77.87±0.61 | 74.53±0.18 | 67.40 |
| Federated | FedProx | ResNet18 | 56.84±0.88 | 62.56±0.87 | 78.33±0.46 | 75.17±0.61 | 68.23 |
| Learning | DRFA | ResNet18 | 56.59±0.34 | 62.87±0.22 | 78.63±0.77 | 75.55±0.42 | 68.41 |
| Methods | FedSR | ResNet18 | 57.56±0.95 | 61.91±0.35 | 78.42±0.19 | 74.73±0.27 | 68.16 |
| | TFL (Ours) | ResNet18 | **59.05**±0.69 | **64.46**±0.21 | **79.35**±0.61 | **76.93**±0.39 | **69.95** |

Table 2: Best ROC-AUCs and corresponding communication round on eICU dataset. This dataset comprises EHRs collected from a diverse range of 72 hospitals across the United States. We trained our model using data from 58 hospitals located in the MIDWEST, NORTHEAST, and SOUTH regions. We evaluated the performance of the global model on an independent set of 14 hospitals from the WEST region. The reported numbers are from three independent runs. *Our approach demonstrates remarkable improvements in OOF robustness with minimal communication costs*.

| Centralized Method | Federated Learning Methods | | | | | | | | | |
|---|---|---|---|---|---|---|---|---|---|---|
| ERM | FedAvg | | FedProx | | DRFA | | FedSR | | TFL (Ours) | |
| | ROC-AUC | # round | ROC-AUC | # round | ROC-AUC | # round | ROC-AUC | # round | ROC-AUC | # round |
| 67.04±1.88 | 57.18±0.03 | 6 | 57.21±0.01 | 6 | 57.20±0.09 | 2 | 57.25±0.03 | 8 | **58.41±0.06** | **2** |

## 4.3 EVALUATION ON SCALABILITY

**Takeaway 2: Learning client topology is communication efficient.** Here, we investigate whether TFL will significantly increase the communication overhead, thus compromising its scalability to large-scale settings. We show the ROC AUC versus communication rounds of the eICU dataset in Figure 7. Our method is found to deliver the highest ROC AUC score while requiring the

Table 3: DSC (↑) score on the FeTS. This dataset contains tumor images from 21 institutions world-wide. We conduct training on 15 institutions and evaluate the model on the remaining 5. The reported numbers are from three independent runs. *Our method delivers the best OOF robustness.*

| Centralized Method | Federated Learning Methods | | | | |
|---|---|---|---|---|---|
| ERM | FedAvg | FedProx | DRFA | FedSR | TFL (Ours) |
| 83.14±0.98 | 71.45±0.05 | 71.15±0.04 | 72.12±1.03 | 72.85±0.05 | **74.29±0.15** |

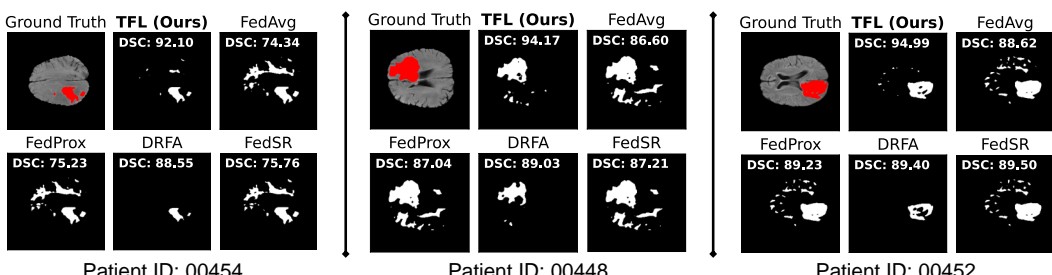

Figure 6: Qualitative results comparison on unseen patients of the FeTS dataset. We show both the tumor segmentation and DSC (↑) score. *Our approach yields consistently superior OOF robustness under diverse local demographics.*

fewest communication rounds, thereby indicating its superior communication efficiency. Interestingly, DRFA also achieves its peak performance within the same number of communication rounds as our method. This is primarily attributable to the fact that both DRFA and our approach utilize a distributionally robust optimization framework. By minimizing the worst-case combination of local client risks, the model is able to converge faster toward the optimum. Additionally, we also show the wall clock time versus OFF accuracy on the PACS dataset in Figure 2. Our method demonstrates communication efficiency approximate to FedAvg and FedSR, yet it delivers superior OOF accuracy.

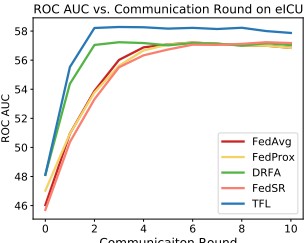

Figure 7: Visualization of ROC AUC vs. comm. round. *Our method yields the best result with fewer communication rounds.*

### 4.4 EVALUATION ON IN-FEDERATION PERFORMANCE

**Takeaway 3: Prioritizing influential clients also benefits in-federation performance.** We have shown that TFL can significantly boost the model's OOF robustness. This naturally leads us to examine its IF performance. We are interested in understanding if TFL can maintain or even improve its IF performance. To empirically evaluate this, we utilized the TPT-48 and CIFAR-10/-100 datasets. TPT-48, which offers additional hold-out data from the IF client, is naturally suitable for assessing IF performance. CIFAR-10 and -100 are widely recognized benchmarks in FL. We show the results in Tables 4 and 5. We observe that our method does indeed bolster the IF performance. For example, TFL outperforms DRFA by $2.2\%$ on TPT-48 and $2.5\%$ on CIFAR-10. These findings underscore the value of focusing on influential clients when building robust models with good IF and OOF performance.

Table 4: Results on TPT-48 dataset. We report results for both IF and OOF evaluations. The reported numbers are the mean of three independent runs. *Our method improves both the OOF and IF performance.*

| Method | Centralized | FedAvg | FedProx | DRFA | TFL (Ours) |
|---|---|---|---|---|---|
| *Out-of-federation* Evaluation (MSE ↓) | | | | | |
| E(24) → W(24) | 0.3998 | 0.6264 | 0.6312 | 0.5451 | **0.4978** |
| N(24) → S(24) | 1.4489 | 2.0172 | 1.9729 | 1.8972 | **1.7432** |
| *In-federation* Evaluation (MSE ↓) | | | | | |
| E(24) → E(24) | 0.1034 | 0.2278 | 0.2163 | 0.1554 | **0.1523** |
| N(24) → N(24) | 0.1329 | 0.1550 | 0.1523 | 0.1437 | **0.1405** |

Table 5: We report results on CIFAR-10/-100. The reported numbers are the mean of three independent runs. *Our method yields the best accuracy at various scales.*

| | CIFAR-10 | | CIFAR-100 | |
|---|---|---|---|---|
| Non-IID | Dir (0.1) | | Dir (0.1) | |
| # of clients | 25 | 50 | 25 | 50 |
| FedAvg | 68.07 | 64.48 | 37.90 | 37.33 |
| FedProx | 68.15 | 64.33 | 37.76 | 37.28 |
| DRFA | 70.51 | 63.19 | 38.04 | 37.54 |
| TFL (Ours) | **72.24** | **65.56** | **38.85** | **38.08** |

## 4.5 ABLATION STUDY

**Effects of partial model approximation.** To handle high-dimensional models, we can leverage partial model parameters to compute the similarity scores for better computation efficiency. Here, we report the OOF performance when using full and partial model parameters. Experiments are conducted on PACS and results are shown in Table 6.

Table 6: TFL's OOF accuracy using full and partial model parameters. We see that *using partial model parameters basically does not affect the performance.*

|         | A          | C          | P          | S          | Average |
|---------|------------|------------|------------|------------|---------|
| Full    | 59.05±0.69 | 64.46±0.21 | 79.35±0.61 | 76.93±0.39 | 69.95   |
| Partial | 58.96±0.85 | 64.57±0.98 | 78.94±0.65 | 76.61±0.13 | 69.77   |

**Effects of graph sparsity.** To avoid learning a fully connected graph with trivial edges, we add a sparsity contain to the client topology step. In our implementation, we adopt the $\epsilon$-graph to make sure Equation 4 is solvable. The threshold value $\epsilon$ controls the graph sparsity. Here, we investigate how it will affect the TFL. We report the results on PCAS in Table 7. We observe that $\epsilon = 0.4$ yields the best performance.

**Effects of client topology update frequency.** The frequency at which the client topology is updated during training affects the overall training time ($f = 50$

Table 7: Ablation on effects of graph sparsity and topology update frequency on PACS.

| Graph Sparsity ($\epsilon$), Accuracy ↑ | | | | |
|-----------------|----------------|-----------------|-----------------|----------------|
| $\epsilon = 0.45$ | $\epsilon = 0.4$ | $\epsilon = 0.38$ | $\epsilon = 0.35$ | $\epsilon = 0.3$ |
| 58.61 | **59.07** | 57.83 | 58.11 | 57.73 |

| Topology update frequency ($f$), Accuracy ↑ | | | | |
|--------|--------|--------|--------|--------|
| $f = 50$ | $f = 30$ | $f = 20$ | $f = 10$ | $f = 5$ |
| **59.19** | 58.91 | 58.27 | 58.80 | 58.28 |

means updating the client topology every 50 rounds). A greater frequency of updates leads to longer running times. We investigated the impact of client topology updating frequency on the performance of the model on PACS. Our results in Table 7 indicate that model performance remains relatively stable across various updating frequencies. We provide more ablation studies in Supplementary D.

## 5 RELATED WORK

**Federated learning.** FL (Li et al., 2020a; Kairouz et al., 2021) has emerged as a powerful tool to protect data privacy in the distributed setting. Current FL methods mainly focus on addressing the in-federation data heterogeneity (Li et al., 2020b; Sattler et al., 2019; Tan et al., 2022). For example, FedProx (Li et al., 2020b), SCAFFOLD (Karimireddy et al., 2020), and FedAlign (Mendieta et al., 2022) tackle this problem from the perspective of learning better generalizable local models. However, limited work discusses the model's out-of-federation (OOF) performance. Orthogonal to existing work, we propose leveraging client relationships to improve the model's OOF generalization capability. **FL generalization to unseen clients.** There are recent attempts to address generalization to unseen clients in FL. FedDG (Liu et al., 2021) is proposed to share the amplitude spectrum of images among local clients to augment the local data distributions. FedADG (Zhang et al., 2021) adopts the federated adversarial training to measure and align the local client distributions to a reference distribution. FedSR (Nguyen et al., 2022) proposes regularizing latent representation's $\ell_2$ norm and class conditional information to enhance the OOF performance. However, existing methods often ignore scalability issues, yielding inferior performance in large-scale distributed setting (Bai et al., 2023). We introduce an approach that employs client topology to achieve good OOD-resiliency in a scalable manner. **Graph topology learning.** The problem of graph topology learning has been studied in data-centralized setting (Mateos et al., 2019; Dong et al., 2019; Stanković et al., 2020; Li et al., 2018; Norcliffe-Brown et al., 2018). However, how to estimate the graph topology with a privacy guarantee has been less investigated. We explore simple methods to infer the graph topology using model weights. We provide a detailed discussion of related work in Supplementary C.

## 6 CONCLUSION

FL faces significant out-of-federation (OOF) generalization challenges that can severely impair model performance. We propose to improve OOF robustness by leveraging client relationships, leading to *Topology-aware Federated Learning* (TFL). TFL contains two key modules: i) Inferring a topology that describes client relationships with model similarity and ii) Leveraging the learned topology to build a robust model. Extensive experiments on real-world and benchmark datasets show that TFL demonstrates superior OOF-resiliency with good communication efficiency.

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

## A    Source Code

The source code for evaluation and visualization can be found in an anonymized repository. Please visit: https://anonymous.4open.science/r/TFL-8390 to access this resource.

## B    Additional Experimental Results

### B.1    Evaluating OOF-resiliency on OfficeHome

We conduct additional experiments on **OfficeHome** (Venkateswara et al., 2017) dataset. It contains $15,588$ images from four domains: art, clipart, product, and real world. The task is a 65-class classification problem. Like PACS's experimental setup, we evenly split each domain into 5 subsets, yielding 20 subsets, and treat each subset as a client. We followed the common *leave-one-domain-out* experiment, where 3 domains are used (15 clients) for training and 1 domain (5 clients) for testing. We use ResNet50 (He et al., 2016) as our model and train the model for 100 communication rounds. Each local client optimized the model using stochastic gradient descent (SGD) with a learning rate of $0.01$, a momentum of $0.9$, weight decay of $5e^{-4}$, and a batch size of $64$. The model is evaluated using classification accuracy.

Table 8: Accuracy on the **OfficeHome** dataset. We conduct experiments using a *leave-one-domain-out* approach, meaning each domain serves as the evaluation domain in turn. Existing methods typically consider each domain as an individual client (Liu et al., 2021; Nguyen et al., 2022). However, in order to simulate a large-scale distributed setting, we took a different approach by further dividing each domain into 5 subsets and treating each subset as a separate client. This increased the total number of clients to 20. *Our method outperformed others across all experimental settings*, demonstrating superior results.

|  | Models | Backbone | OfficeHome | | | | |
|---|---|---|---|---|---|---|---|
|  |  |  | A | C | P | R | Average |
| Centralized | Mixup (Xu et al., 2020) | ResNet50 | 64.7 | 54.7 | 77.3 | 79.2 | 69.0 |
| Methods | CORAL (Sun & Saenko, 2016) | ResNet50 | 64.4 | 55.3 | 76.7 | 77.9 | 68.6 |
| Federated | FedAvg | ResNet50 | 24.10 | 23.16 | 40.19 | 43.47 | 32.73 |
| Learning | FedProx | ResNet50 | 23.16 | 23.47 | 41.08 | 42.66 | 32.59 |
| Methods | DRFA | ResNet50 | 25.29 | 23.98 | 41.23 | 42.35 | 33.21 |
|  | FedSR | ResNet50 | 23.51 | 22.93 | 39.30 | 41.48 | 31.81 |
|  | TFL (Ours) | ResNet50 | **26.37** | **24.47** | **43.96** | **44.74** | **34.89** |

### B.2    Data Pre-processing on eICU

We follow (Huang et al., 2019) to predict patient mortality using drug features. These features pertain to the medications administered to patients during the initial 48 hours of their ICU stay. We've extracted pertinent patient and corresponding drug feature data from two primary sources: the 'medication.csv' and 'patient.csv' files. Our final dataset is a table with the dimension of $19000 \times 1411$. Each row in this matrix symbolizes a unique patient, while each column corresponds to a distinct medication.

### B.3    More Results on eICU

We conduct more experiments on the eICU dataset to evaluate the gap between in-distribution (ID) and out-of-distribution (OOF) and visualize the results in Figure 8. *We observe that existing FL methods are not robust against OOF data.*

## C    Detailed Discussion of Related Work

**Federated learning.** Federated learning (Li et al., 2020a; Kairouz et al., 2021) has emerged as a powerful tool to protect data privacy in the distributed setting. It allows multiple clients/devices to collaborate in training a predictive model without sharing their local data. Despite the success, current FL methods are vulnerable to heterogeneous data (non-IID data) (Li et al., 2020b; Sattler et al., 2019), a common issue in real-world FL. Data heterogeneity posits significant challenges

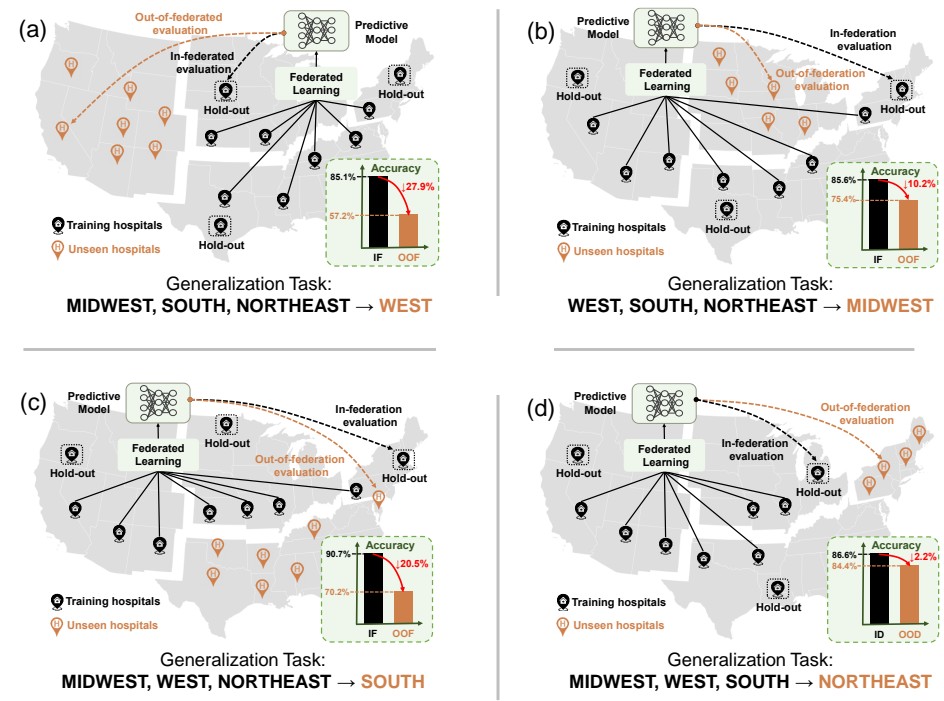

Figure 8: Additional results on eICU (ID *vs.* OOF performance). eICU's 72 hospitals are distributed across the United States. Specifically, there are 14 hospitals in the WEST, 28 hospitals in the MIDWEST, 26 hospitals in the SOUTH, and 4 hospitals in the NORTHEAST. We employ a *leave-one-region-out* approach, designating one geographic region as the OOF region while the remaining as ID regions. We observe a considerable gap between ID and OOF performance, indicating that *current FL methods are not robust against OOF data.*

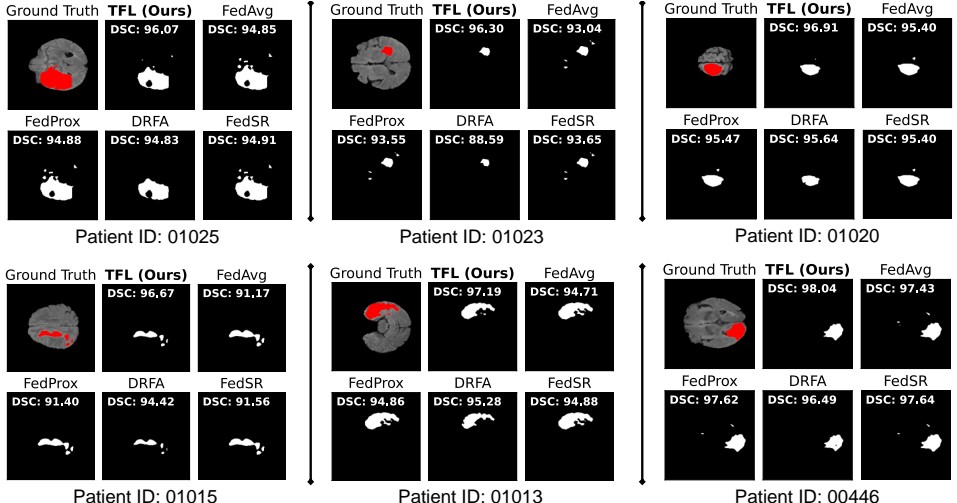

Figure 9: Additional qualitative results comparison on unseen patients of the FeTS dataset. We show both the tumor segmentation and DSC (↑) score. *Our method demonstrates consistent superior OOF-resiliency across a range of local demographics.*

to FL, such as the severe convergence issue (Li et al., 2020b) and poor generalization ability to new clients (Sattler et al., 2019). To improve the model's robustness against data heterogeneity, `FedProx` add a proximal term to restrict the local model updating, avoiding biased models toward local data distribution. `SCAFFOLD` (Karimireddy et al., 2020) introduces a control variate to rectify

the local update. `FedAlign` (Mendieta et al., 2022) improve the heterogeneous robustness by training local models with better generalization ability. However, most FL methods focus on the model's in-distribution performance. Orthogonal to existing work, we propose leveraging client relationships to improve the model's OOF generalization capability.

**FL generalization to unseen clients.** A handful of works tackle generalization to unseen clients in the FL setting. FedDG (Liu et al., 2021) is proposed to solve domain generalization in medical image classification. The key idea is to share the amplitude spectrum of images among local clients to augment the local data distributions. FedADG (Zhang et al., 2021) adopts the federated adversarial training to measure and align the local client distributions to a reference distribution. FedGMA (Tenison et al., 2022) proposes gradient masking averaging to prioritize gradients aligned with the overall domain direction across clients. FedSR (Nguyen et al., 2022) proposes regularizing latent representation's $\ell_2$ norm and class conditional information to enhance the OOF performance. However, existing methods often ignore scalability issues, yielding inferior performance in large-scale distributed setting (Bai et al., 2023). In this paper, we introduce an approach that employs client topology to strike a good balance between OOF-resiliency and scalability.

**Graph topology learning.** The problem of graph topology learning has been studied in different fields. In graph signal processing (Mateos et al., 2019; Dong et al., 2019; Stanković et al., 2020), existing work explore various way to learn the graph structure from data with structural regularization ( *e.g.,* sparsity, smoothness, and community preservation (Zhu et al., 2021)). In Graph Neural Networks (GNNs) (Wu et al., 2020; Welling & Kipf, 2016), researchers have explored scenarios where the initial graph structure is unavailable, wherein a graph has to be estimated from objectives (Li et al., 2018; Norcliffe-Brown et al., 2018) or words (Chen et al., 2019; 2020b). The existing graph topology learning methods often require centralizing the data, making it inapplicable in federated learning. However, how to estimate the graph topology with a privacy guarantee has been less investigated. In this paper, we explore simple methods to infer the graph topology using non-private information, *i.e.,* model weights.

## D   ADDITIONAL ABLATION STUDY

**Hyperparameter** $q$**.** We investigate the impact of hyperparameter $\eta$ on eICU. Our findings demonstrate that setting $q = 0.1$ yields the best results. **Centrality.** We employed betweenness centrality to derive the topological prior. However, it is worth noting that other types of centrality, such as degree (Freeman et al., 2002) and closeness (Bavelas, 1950), could also be utilized. We conducted experiments on eICU to verify the impact of different centrality measures on TFL. Our findings indicate that betweenness centrality produces the best result. **Similarity metrics.**

Table 9: Ablation study evaluating the efficacy of hyperparameter tuning, centrality., and similarity metric.

| Effectiveness of $q$, ROC AUC ↑ | | | | |
|---|---|---|---|---|
| $q = 1.0$ | $q = 1e^{-1}$ | $q = 1e^{-2}$ | $q = 1e^{-3}$ | $q = 1e^{-4}$ |
| 57.91 | **58.31** | 57.43 | 56.96 | 57.29 |
| Effectiveness of centrality, ROC AUC ↑ | | | | |
| Betweenness | Degree | Closeness | Eigenvector | Current flow |
| **58.28** | 57.69 | 57.86 | 57.57 | 57.83 |
| Effectiveness of similarity measure, Accuracy ↑ | | | | |
| | $\ell_1$ | $\ell_2$ | dot produt | cosin |
| OOF Accuracy | 58.11 | 58.26 | 59.14 | 58.52 |

We investigate how the model performs under different similarity metrics on PACS. We found that the dot product-based metric produces the best results.

## E   IMPLEMENTATION DETAILS

**Experiment settings and evaluation metrics.** For the **PACS dataset**, we evenly split each domain into 5 subsets, yielding 20 subsets, and we treat each subset as a client. We followed the common "leave-one-domain-out" experiment, where 3 domains are used (15 clients) for training and 1 domain (5 clients) for testing. We evaluated the model's performance using classification accuracy. We use ResNet18 (He et al., 2016) as our model and train the model for 100 communication rounds. Each local client optimized the model using stochastic gradient descent (SGD) with a learning rate of $0.01$, momentum of $0.9$, weight decay of $5e^{-4}$, and a batch size of $8$. For **CIFAR-10/100**, we adopt the same model architecture as FedAvg (McMahan et al., 2017). The model has 2 convolution layers with $32, 64$ $5 \times 5$ kernels, and 2 fully connected layers with $512$ hidden units. we use Dirichlet

---

**Algorithm 1** Topology-aware Federated Learning

---

**Input:** $K$ clients; learning rate $\eta_\theta$ and $\eta_\lambda$; communication round $T$; initial model $\theta^{(0)}$; initial $\lambda^{(0)}$; topology update frequency $f$ .
**while** not convergence **do**
    **for** each communication round $t = 1, \cdots T$ **do**
        server **samples** $m$ clients according to $\boldsymbol{\lambda^{(t)}}$
        **for** each client $i = 1, \cdots m$ in parallel **do**
            $\theta_i^{t+1} = \theta_i^t - \eta_{\theta_i^t} \nabla_{\theta_i^t} F(\theta_i^t)$
            client $i$ send $\theta_i^{t+1}$ back to the server
        **end for**
        server **computes** $\boldsymbol{\theta}^{t+1} = \sum_{i=1}^m \theta_i^{t+1}$
        **if** $t \% f == 0$ **then**
            Updating graph $\mathcal{G}$ via Equation 4
        **end if**
        calculating topological prior $p$ from $\mathcal{G}$
        calculating $\nabla_{\lambda^{(t)}} F(\boldsymbol{\theta}^{(t+1)}, \boldsymbol{\lambda^{(t)}})$ via Equation 6
        $\boldsymbol{\lambda}^{t+1} = \mathcal{P}_{\Delta_K}(\boldsymbol{\lambda}^t + \eta_\lambda^t \nabla_{\lambda^{(t)}} F(\boldsymbol{\theta}^{(t+1)}, \boldsymbol{\lambda^{(t)}}))$
    **end for**
**end while**

---

distribution (Hsu et al., 2019) to partition the dataset into the heterogeneous setting with 25 and 50 clients. For the **eICU dataset**, we treat each hospital as a client. We use a network of three fully connected layers. This architecture is similar to (Huang et al., 2019; Sheikhalishahi et al., 2020). We train our model for 30 communication rounds, using a batch size of 64 and a learning rate of 0.01, and report the performance on unseen hospitals. Within each communication round, clients performs 5 epochs (E = 5) of local optimization using SGD. The evaluation metric employed was the ROC-AUC, a common practice in eICU (Huang et al., 2019). For the **FeTS dataset**, we treat each institution as a client. We adopt the widely used U-Net (Ronneberger et al., 2015) model. We train our model for 20 communication rounds, using a learning rate of 0.01 and a batch size of 64. We conduct training with 16 intuitions and report results on 5 unseen institutions. Each institution performs 2 epochs of local optimization (E = 2) using SGD. The evaluation metric is Dice Similarity Coefficient (DSC ↑). For **TPT-48**, we consider two generalization tasks: (1) E(24) → W(24): Using the 24 eastern states as IF clients and the 24 western states as OOF clients; (2) N(24) → S(24): Using the 24 northern states as IF clients and the 24 southern states as OOF clients. We use a model similar to (Xu et al., 2022), which has 8 fully connected layers with 512 hidden units. We use SGD optimizer with a fixed momentum of 0.9. The evaluation metric is Mean Squared Error (MSE ↓). Algorithm 1 shows the overall algorithm of TFL. In implementation, we used dot product as the metric to measure client similarity.

## F  DISCUSSION OF LIMITATIONS

In this section, we discuss the limitations of TFL and the potential solutions.

**Concerns on privacy leakage.** Client topology learning may raise concerns about (unintentional) privacy leakage. However, we argue that any such leakage would be a general issue for FL methods rather than a unique concern for our approach. In comparison to standard FL, our method does not require additional information to construct the client topology, thus providing no worse privacy guarantees than well-established methods like FedAvg (McMahan et al., 2017) and FedProx (Li et al., 2020b). Nonetheless, FL may still be vulnerable to attacks that aim to extract sensitive information (Bhowmick et al., 2018; Melis et al., 2019). In future work, we plan to explore methods for mitigating (unintentional) privacy leakage.

**Concerns on high-dimensional node embedding.** As outlined in Section 3.1, we harness model weights as node embeddings. Nevertheless, incorporating large-scale models, such as Transformers (Vaswani et al., 2017; Dosovitskiy et al., 2021), may present a formidable obstacle, producing an overwhelmingly high-dimensional node vector. This will significantly increase computational demands for assessing node similarity. We argue that this can be addressed by dimension reduction.

There are two possible ways: ① Utilizing model weights of certain layers as node embedding instead of the whole model. ② Directly learning the low-dimensional node embedding. One simple idea is to leverage Hypernetworks (Shamsian et al., 2021) to learn the node embedding with controllable dimensions.

**Concerns on high computation cost for cross-device FL.** Client topology learning requiris $\mathcal{O}(N^2)$ computation complexity for $N$ clients. This quadratic complexity is prohibitively expensive in cross-device FL, where hundreds, thousands, or millions of clients/devices may be involved. In this case, we argue that the computation cost can be significantly reduced by client clustering (Sattler et al., 2020; Ghosh et al., 2020). By partitioning the clients into clusters, the total number of "clients" is reduced, allowing for cluster-level client topology learning to estimate the topology with reduced computation costs. We conducted experiments on the eICU dataset to empirically validate the effectiveness of our clustering-based method. The eICU dataset was selected for its large scale (72 clients) compared to all other evaluated datasets. Specifically, during client topology learning, we use KMeans (Lloyd, 1982) to partition the training clients into several (*e.g,* 10) clusters and learn the client topology at the cluster level. As shown in Table 10, our clustering approach significantly reduces computation costs by 69%, with only a small decrease in OOF performance by 0.77%.

Table 10: Compassion of computation of wall-clock time on eICU dataset. *Our clustering approach significantly reduces computation costs by* 69%*, with only a small decrease in OOF performance by* 0.77%.

|  | ROC-AUC | Wall-clock time (s) |
|---|---|---|
| FedAvg | 57.18 ± 0.03 | 120.15 |
| TFL | 58.41 ± 0.06 | 437.61 |
| TFL w/ Clustering | 57.96 ± 0.18 | 133.08 |

