# OpenReview forum: "Towards Out-of-federation Generalization in Federated Learning"
_ICLR.cc/2024/Conference — ICLR 2024 Conference Desk Rejected Submission_

### Official Review · Reviewer_RF1L · 2023-10-31

**Soundness:** 2 fair
**Presentation:** 3 good
**Contribution:** 2 fair
**Rating:** 6
**Confidence:** 4

**Summary:**

The paper considers the problem of out-of-federation generalization in the context of federated learning. The goal is to train a model able to generalize well to unseen clients during training. In order to achieve this goal, the paper proposes Topology-aware Federated Learning (TFL), a framework combining two ideas: robust/agnostic federated learning (Deng et al., 2020; Mohri et al., 2019), and federated multi-task learning (Smith et al., 2017; Vanhaesebrouck et al., 2017).

TFL considers an optimization problem over the model parameters and importance of each client, as well as the client topology capturing the clients' relationship. In order to solve this optimization problem, TFL acts iteratively: at each iteration, the  model parameters and importance of each client are optimized for a fixed topology, then the topology is updated for fixed model parameters and importance vector.

The paper curates two out-of-federation benchmarks using real-world healthcare data, and empirically evaluates the performance of TFL on these and standard benchmarks.

**Strengths:**

- The paper curates two out-of-federation benchmarks using real-world healthcare data. Both datasets could be beneficial in future work.
- The numerical experiments are extensive and covers many aspects of the proposed learning framework.

**Weaknesses:**

- The technical novelty of the paper is limited. The paper simply combines the ideas from  robust/agnostic federated learning (Deng et al., 2020; Mohri et al., 2019), and federated multi-task learning (Smith et al., 2017; Vanhaesebrouck et al., 2017).
- The notation and the theoretical results are not rigorous:
    - The definition of $\Theta$ is not consistent when moving from (2) to (3). In (2), we optimize the parameters of one global model, while in (3), we optimizer the individual parameters of each client.
    - The function $F$ as defined in (6) ought to be iteration-dependent since $\mathbf{p}$ has the potential to vary from one iteration to another.
    - The function $F$ is defined in two different manners in (3) and (6). The function $F$ as defined in (3) should depend on $W$, while the function $F$ defined in (6) should depend on $\mathbf{p}$.
    - In light of the previous point, it is unclear what is the statement of Theorem 1.
    - I am uncertain about how to interpret Theorem 2.
- Other minor issues:
    - In the abstract and introduction, the paper conveys an initial emphasis on healthcare data. However, I believe it would be more effective if the paper avoids an exclusive focus on healthcare data, considering that the proposed approach holds broader applicability. Instead, I recommend the authors highlight healthcare as one potential use case rather than positioning it as the sole focus of the paper.
    - In Figure 2, it is unclear why FedProx need more communication in comparison with FedAvg.
    - In the opening of Section 2, the paper asserts that the empirical risk is equal to the population risk, a statement generally incorrect. I believe the authors intended to convey that the population risk is approximately equal to the empirical risk.

**Questions:**

I am of the opinion that the current theoretical results are not entirely accurate. Should the authors fail to refute my assertion, I recommend considering the removal of these theoretical results from the paper. However, if the authors are able to either demonstrate the correctness or agree to exclude the theoretical results, I am open to revising my evaluation positively.

---

> ### Author Response · Authors · 2023-11-22
> **Response to Reviewer RF1L**
>
> Q1. **Clarification on technical novelty**
> > The technical novelty of the paper is limited. The paper simply combines the ideas from robust/agnostic federated learning (Deng et al., 2020; Mohri et al., 2019), and federated multi-task learning (Smith et al., 2017; Vanhaesebrouck et al., 2017).
>
> Our method is NOT a simple combination of existing ideas; it is significantly different in tackling the Out-Of-Federation (OOF) generalization problem. **First**, Directly applying robust/agnostic FL to tackle OOF generalization may suffer from the over-pessimism problem (GroupDRO, Sagawa et al., 2019; Data geometry, Liu et al., 2022), leading to poor OOF-resiliency (as discussed in Sec. 2). To this end, we propose to minimize the model risks over not only worst-case but also the influential clients, thus building a more balanced model with better OOF robustness (Verified via extensive empirical results). **Second**, how to identify influential clients for the OOF scenario is largely underexplored. Federated multi-task learning does not have a discussion on how to identify influential clients. Our method innovatively employs client centrality as a measure of influence. And we have developed a technique to seamlessly integrate client’s influential information into the robust optimization process.
>
> Q2. **On notation and the theoretical results**
>
> We have refined the notations in our paper to improve clarity and consistency. The following are the key changes we have implemented: (i) In equation (3), we introduced a dependency on W to function F. Additionally, we adjusted the second term to highlight its role in measuring the similarity between client k and client j. How to calculate this term is deferred to Equation (4). All modifications ensure that equation (3) is consistent with equation (2). (ii) We recognized the necessity for equation (6) to be dependent on p. Consequently, p has been incorporated into function F.
>
> Regarding the theoretical results, we acknowledge that more work is needed to refine them. We will remove it from the appendix. However, we are committed to continuing our exploration of the theoretical analysis to enhance its clarity.
>
> Q3. **Other minor issues**
> > In the abstract and introduction, the paper conveys an initial emphasis on healthcare data. However, I believe it would be more effective if the paper avoids an exclusive focus on healthcare data, considering that the proposed approach holds broader applicability. Instead, I recommend the authors highlight healthcare as one potential use case rather than positioning it as the sole focus of the paper.
>
> The reviewer’s feedback is constructive and can broaden the audience of this paper. We have adjusted the abstract and introduction correspondingly by highlighting not exclusively focusing on healthcare.
>
> > In Figure 2, it is unclear why FedProx needs more communication in comparison with FedAvg.
>
> Figure 2 shows the wall-clock time and the optimal performance of each method. It's important to note that wall-clock time is affected by both communication and computation costs. In the case of FedProx, there is an increased amount of local computation involved. This additional computation contributes to a higher wall-clock time for FedProx when compared to FedAvg.
>
> > In the opening of Section 2, the paper asserts that the empirical risk is equal to the population risk, a statement generally incorrect. I believe the authors intended to convey that the population risk is approximately equal to the empirical risk.
>
> Thanks for the reviewer’s comments. We have adjusted the sentences accordingly in the paper (highlighted in light blue).

---

> > ### Comment · Reviewer_RF1L · 2023-11-22
> >
> > I thank the authors for their rebuttal. The provided response aligns with my expectations, prompting me to raise my score.

---

> > > ### Author Response · Authors · 2023-11-22
> > > **Thank you for your prompt reply**
> > >
> > > We sincerely appreciate the valuable insights and constructive feedback provided by the reviewer. Your thoughtful comments have significantly contributed to enhancing the overall quality of our paper.

---

### Official Review · Reviewer_L1rQ · 2023-10-31

**Soundness:** 3 good
**Presentation:** 3 good
**Contribution:** 3 good
**Rating:** 6
**Confidence:** 3

**Summary:**

In this paper, the authors proposed a new optimization framework for solving out-of-federation (OOF) problems in federated learning. In particular, they proposed to alternatively optimize a client graph topology and minimize the overall weighted loss whose weights closely dependent on the graph topology. In this case, the new optimization framework can leverage the influential clients and also the “outliers”. They conducted comprehensive numerical experiments to compare the proposed framework with several existing baselines on both real-world datasets and some FL benchmark datasets. The proposed framework has marginal improvements over existing baselines.

**Strengths:**

Overall, the paper is clear and easy to understand. The proposed framework seems novel. The authors conducted many empirical experiments to demonstrate the performance of the proposed framework.

**Weaknesses:**

1. As the authors has mentioned, solving the client topology learning problem requires $O(N^2)$, which does not scale well when $N$ is large. Although the authors provide an alternative solution: clustering based method, no experiments has conducted under such scenarios.
2. In the existing literature, there are a few papers have discussed how to measure the similarity between two clients’ local distribution, for example, using prototype model. How's the method used in this paper compared with those ones?

> Tan, Y., Long, G., Liu, L., Zhou, T., Lu, Q., Jiang, J., & Zhang, C. (2021). FedProto: Federated Prototype Learning across Heterogeneous Clients. AAAI Conference on Artificial Intelligence.

3. I wonder if the proposed framework still works when there is/are an/some adversarial client(s) presented in the FL system. Sometimes the adversarial behavior may occur due to connectivity issues.

**Questions:**

1. Page 4, line 4, “In the objective function, the first term follows the same spirit of Equation 3…”, a typo? Is this Equation 2?
2. Figure 4, do clients use the same training algorithm and same training hyperparameters? If the clients uses different algorithm or training hyperparameters, I doubt their model parameters will be similar even if they have similar local data distribution.
3. In Equation 4, why using cosine similarity and $\ell_0$ distance? In figure 4, it seems that other differentiable  distances also follow the same trend as cosine similarity.

---

> ### Author Response · Authors · 2023-11-22
> **Response to Reviewer L1rQ**
>
> Q1. **Experiment on TFL with client clustering**
> > As the authors has mentioned, solving the client topology learning problem requires $O(N^2)$, which does not scale well when
> $N$ is large. Although the authors provide an alternative solution: clustering based method, no experiments has conducted under such scenarios.
>
> We conducted experiments on the eICU dataset to empirically validate the effectiveness of our clustering-based method. The eICU dataset was selected for its large scale compared to all other evaluated datasets. As shown in the following table, our clustering approach significantly reduces computation costs by 69%, with only a small decrease in OOF performance by 0.77%.
>
> |                   |     ROC-AUC    | Wall-clock time (s) |
> |:-----------------:|:------------:|:---------------:|
> |       FedAvg      | 57.18±0.03 |      120.15     |
> |        TFL        | 58.41±0.06 |      437.61     |
> | TFL w/ Clustering | 57.96±0.18 |      133.08     |
>
> Q2. **Compared with prototype-based method (FedProto)**
> > In the existing literature, there are a few papers have discussed how to measure the similarity between two clients’ local distribution, for example, using prototype model. How's the method used in this paper compared with those ones?
>
> The prototype-based method offers an alternative approach to assessing client similarity. However, there are two key distinctions between this method and ours. First, in contrast to prototype-based methods that are specifically designed for classification. Our method is more general and can easily handle various tasks (classification, regression, and segmentation; we empirically verified all three tasks). Second, even for classification tasks, the prototype-based method needs to learn additional prototypes for each class. Our method, on the other hand, does not require this extra step, making the training process more efficient. We have revised the related work to have a discussion on the prototype method and cite the mentioned paper.
>
> Q3. **Discussion on adversarial client(s)**
> > I wonder if the proposed framework still works when there is/are an/some adversarial client(s) presented in the FL system. Sometimes the adversarial behavior may occur due to connectivity issues.
>
> We see the potential of using client topology to tackle adversarial clients. In scenarios where clients, due to system or network failures, disobey the protocols and send arbitrary messages (such as shuffled, sign-flipped, or noised parameters), our client topology learning approach becomes particularly useful. These adversarial behaviors typically result in models that show lower similarity to normal models. By leveraging client topology learning, we can identify these adversarial clients as isolated nodes within the topology.
>
> Once identified, applying centrality measures to these nodes can effectively lower their importance scores. This approach minimizes their impact on the overall model aggregation process. Therefore, we believe our method possesses a degree of robustness against adversarial scenarios in FL. Improving FL’s adversial robustness is an important and interesting problem, we will leave the exploration of TFL in this direction to future work.
>
> Q4. **Minors**
> > Page 4, line 4, “In the objective function, the first term follows the same spirit of Equation 3…”, a typo? Is this Equation 2?
>
> Yes, this is a typo. We have adjusted it correspondingly in the paper.
>
> > Figure 4, do clients use the same training algorithm and same training hyperparameters?
>
> All the models are trained using the same training algorithm and hyperparameters. We have adjusted the corresponding sentence to make it clear.
>
> > In Equation 4, why using cosine similarity and $l_0$ distance? In figure 4, it seems that other differentiable distances also follow the same trend as cosine similarity.
>
> In Equation 4, "sim" denotes any similarity measure. $l_0$ norm is used to enforce the sparsity of the client topology.

---

> > ### Author Response · Authors · 2023-11-22
> > **We are keen to discuss further with you**
> >
> > Dear Reviewer **L1rQ**,
> >
> > We thank you again for your time and your constructive comments.
> >
> > We would really appreciate it if you could kindly let us know whether there are any further questions. We will be more than happy to address them.
> >
> > Best wishes,
> >
> > Authors

---

### Official Review · Reviewer_Q4J6 · 2023-11-01

**Soundness:** 3 good
**Presentation:** 3 good
**Contribution:** 3 good
**Rating:** 6
**Confidence:** 3

**Summary:**

This paper studies the OOF generalization problem in federated learning, i.e., whether a trained global model can generalize to new clients that do not participate in FL training. The author propose a method to construct a client similarity graph, and emphasize those “influential” clients in the graph. The algorithm is empirically shown to be effective and outperforms a line of federated DG baselines.

**Strengths:**

1. The paper is well-written and easy to follow. The algorithm is clear.
2. The experiments are extensive and verify the superior performance of the proposed algorithm.

**Weaknesses:**

1. A detailed explanation of the motivation of the algorithm will be beneficial. For example, what is the motivation behind emphasizing influential client? Is it because the OOF clients’ distribution are more likely to be similar to these clients? If so, why is that the case? If not, why up-weighting these clients?
2. The author claim that solving Equation 4 with $l_0$ can be NP-hard. However, it seems to be wrong. In this objective function, the optimization for each $w_{k, l}$ is purely disentangled, since $\\|W\\|\_0 = \sum_{k, l} 1\\{w\_{k, l} > 0\\}$. And the solution is just very similar to the proposed method, if $\|W\|_0$ is weighted by epsilon. This does not hurt the soundness of the method.

**Questions:**

1. How the wall-clock time is calculated in Figure 2? Usually, wall-clock time is influence by both computation and communication cost, and their weights depend on the bandwidth, delay, device, … I believe number of communication rounds or total # bits transmitted could be a better metric for communication efficiency.

Minor: page 4 line 4: Equation 3 -> Equation 2

---

> ### Author Response · Authors · 2023-11-22
> **Response to Reviewer Q4J6**
>
> Q1. **More detailed explanation**
> > A detailed explanation of the motivation of the algorithm will be beneficial. For example, what is the motivation behind emphasizing influential clients? Is it because the OOF clients’ distribution is more likely to be similar to these clients? If so, why is that the case? If not, why up-weighting these clients?
>
> We appreciate the reviewer's suggestion and have revised the second paragraph on page 2 to provide a more detailed explanation for emphasizing influential clients. Our rationale is based on the idea that influential clients are the most representative within the federation. Therefore, they are more likely to be similar to unseen OOF clients. Then we use a concrete example from healthcare to support our claim.
>
> Q2. **The sparsity term in Equation 4**
>
> Thanks for pointing out. We have adjusted the sentences accordingly in the paper (highlighted in light blue).
>
> Q3. **On wall-clock time**
> > How the wall-clock time is calculated in Figure 2? Usually, wall-clock time is influence by both computation and communication cost, and their weights depend on the bandwidth, delay, device, … I believe number of communication rounds or total # bits transmitted could be a better metric for communication efficiency.
>
> The wall-clock time is calculated by running each method individually on the same machine until optimal performance is achieved. We have different opinions on choosing communication round over wall-clock time as the metric. The scalability of FL can also be affected by the computation complexity. So, wall-clock time could be a holistic metric to assess scalability. We have adjusted the corresponding to make this point clear. Additionally, in the experimental section (Figure 7), we provide a comparison of the methods' efficiency in terms of communication rounds. The following table shows the optimal performance alongside the total number of communicated parameters. Our method achieves the best performance with the fewest total communicated parameters.
>
> |            | ROC-AUC | Total # of comm. params |
> |:----------:|:-----:|:-----------------------:|
> |   FedAvg   | 57.18 |          15.66M         |
> |   FedProx  | 57.21 |          15.66M         |
> |    DRFA    | 57.20 |          10.44M         |
> |    FedSR   | 57.25 |          20.88M         |
> | TFL (Ours) | 58.41 |          10.44M         |

---

> > ### Author Response · Authors · 2023-11-22
> > **We are keen to discuss further with you**
> >
> > Dear Reviewer **Q4J6**,
> >
> > We thank you again for your time and your constructive comments.
> >
> > We would really appreciate it if you could kindly let us know whether there are any further questions. We will be more than happy to address them.
> >
> > Best wishes,
> >
> > Authors

---

> > > ### Comment · Reviewer_Q4J6 · 2023-11-22
> > > **Thanks!**
> > >
> > > Thanks for your rebuttal.
> > >
> > > Regarding the wall-clock time, I still believe running each method on single machine and collecting the wall-clock time may not be an appropriate way to measure efficiency, especially given the facts that (1) real FL systems involves multiple edge devices, even for cross-silo FL, and more importantly (2) in real FL systems, it is the **communication cost** that dominate. In the FedAvg paper, it is claimed that "in federated optimization communication costs dominate". However, recording the wall-clock time on single machine mainly just reflect the computational cost, not the communication cost. Please correct me if you are actually running the experiments with multiple machines.
> > >
> > > However, I appreciate the additional table regarding the total # of communication parameters. Could you please provide insights on why your TFL requires less parameters to communicate? Thanks!

---

> > > > ### Author Response · Authors · 2023-11-22
> > > > **Response to Reviewer Q4J6**
> > > >
> > > > We thank the reviewer for the prompt reply and insightful discussion on communication overhead. Due to hardware resource constraints, we can only follow the practice of academic FL to run simulations on a single machine. In this case, we believe the total # of communication parameters demonstrates the efficiency of our method.
> > > >
> > > > Our insight: TFL actively samples influential and worst-case clients that are more informative than random sampling to participate in each round. This can reduce the unnecessary redundancy in the randomly sampled clients, thereby improving the convergence. This faster convergence reduces the number of rounds needed, consequently decreasing the total communicated parameters.

---

### Official Review · Reviewer_Nx6V · 2023-11-03

**Soundness:** 2 fair
**Presentation:** 2 fair
**Contribution:** 2 fair
**Rating:** 5
**Confidence:** 4

**Summary:**

This work tackles the problem of out-of-generalization in FL where the trained model from conventional FL performs poorly for clients outside of the current federation with different distributions. The work proposes to leverage client topology where the relationships across the clients are learned with a weight matrix as we do for graphs. The relationships are learned with pair-wise similarity with only the last few layers of the clients' models. The authors claim that this is communication efficient. The authors include experimental results on a variety of datasets and compare the performance with baselines such as FedAvg, FedProx, DRFA.

**Strengths:**

- The work investigates a relevant problem in FL where clients with local data coming from data distributions different from which the global model is trained on suffer from bad performance.

- The work builds upon previous work on graph centrality to propose client centrality and topology for improving previous DRFL methods.

- The work provides experimental validation on extensive number of datasets and different baselines.

**Weaknesses:**

- A main concern I have is regarding the part where we have to learn the client topology through the similarity measures in eq. (3). This requires training over all of the clients' models (even if it is some of the last layers) which can incur large computation overhead for cross-device FL scenarios where the number of clients can easily range to millions of clients. While the authors argue that models can be freely shared among clients and this can address privacy issues, I am unsure why this can ensure privacy. I think it will do the opposite.

- Another concern I have is regarding eq.(5) where the authors try to figure out the influential clients that represent the distribution of out-of-federation clients, along with the worst distribution clients. Wouldn't this lead to potential bias to the distribution of the influential clients? Let us assume there is a setting where there are mainly three groups of clients with different distributions within the out-of-federation group. Wouldn't this lead to the algorithm biasing the model towards one of the distribution and not performing well for the other two groups?

- Lastly, regarding the experimental results, the authors argue that the method is communication efficient since the achieved targeted performance occurs in an earlier communication round for the proposed method compared to other methods. However, I wonder if this is actually the case for the actually communicated number of parameters (for example in Figure7)?

Overall due to these concerns I am leaning towards rejection, but I look forward to the discussions with other reviewers and authors.

**Questions:**

See weaknesses above.

---

> ### Author Response · Authors · 2023-11-22
> **Response to Reviewer Nx6V (1/2)**
>
> Q1. **Computation overhead**
> > A main concern I have is regarding the part where we have to learn the client topology through the similarity measures in eq. (3). This requires training over all of the clients' models (even if it is some of the last layers) which can incur large computation overhead for cross-device FL scenarios where the number of clients can easily range to millions of clients.
>
> We thank the reviewer for pointing out the concern we already discussed in the method section (Page 5 Discussion). In our discussion, we mention that the computation costs for cross-device FL can be significantly reduced via client clustering. By partitioning the clients into clusters, the total number of “clients” is reduced, allowing for cluster-level client topology learning with reduced computation costs.
>
> We conducted experiments on the eICU dataset to empirically validate the effectiveness of our clustering-based method. The eICU dataset was selected for its large scale (72 clients) compared to all other evaluated datasets. As shown in the following table, our clustering approach significantly reduces computation costs by 69%, with only a small decrease in OOF performance by 0.77%. For more details please refer to the supplementary Section F.
>
> |                   |     ROC-AUC    | Wall-clock time (s) |
> |:-----------------:|:------------:|:---------------:|
> |       FedAvg      | 57.18±0.03 |      120.15     |
> |        TFL        | 58.41±0.06 |      437.61     |
> | TFL w/ Clustering | 57.96±0.18 |      133.08     |
>
> Q2. **Privacy issue**
> > While the authors argue that models can be freely shared among clients and this can address privacy issues, I am unsure why this can ensure privacy. I think it will do the opposite.
>
>  **Why this can ensure privacy**: As discussed in Sec. 3.1, traditional topology learning methods generally require access to the data, violating the privacy constraint. We follow the same spirit of FL to learn client topology from model parameters to protect data privacy.
>
> **Information leakage by model sharing**: Sharing model parameters is common practice in FL (Tian et al., 2020; McMahan et al., 2017). Our primary goal is to ensure basic privacy protection in client topology learning. Further discussions on (unintended) information leakage of sharing models (a fundamental problem for FL (Zhu et al., 2019) are out of the scope of this paper. To prevent any misunderstandings, we have revised the relevant sentences to ensure better clarity.
>
> Q3. **Clarification on eq.(5)**
> > Another concern I have is regarding eq.(5) where the authors try to figure out the influential clients that represent the distribution of out-of-federation clients, along with the worst distribution clients. Wouldn't this lead to potential bias to the distribution of the influential clients? Let us assume there is a setting where there are mainly three groups of clients with different distributions within the out-of-federation group. Wouldn't this lead to the algorithm biasing the model towards one of the distribution and not performing well for the other two groups?
>
> The reviewer’s understanding of eq.(5) seems incorrect. We would like to clarify a couple of points to address your concerns: **First**, Influential clients are identified prior to the optimization process of eq. (5), not during it. This means we first establish who the influential clients are using the learned client topology. Once identified, eq.(5) leverages the client's influence as prior knowledge to guide the model optimization. This process is iterative – the models are updated based on eq. (5), and then the influential clients are re-evaluated based on these updated models. **Second**, eq.(5) will not only prioritize influential clients. Instead, it will minimize risks over both worst-case and influential clients. By choosing proper \tau, eq.(5) will build balanced models by striking a good tradeoff between worst-case and influential clients. Thus, it will not lead to models biased toward influential clients. In Sec. 3.2, we provided a detailed justification of our design choice from the perspective of the uncertain set (Hu et al., 2018; Frogner et al., 2021).

---

> ### Author Response · Authors · 2023-11-22
> **Response to Reviewer Nx6V (2/2)**
>
> Q4. **On communication efficiency**
> > Lastly, regarding the experimental results, the authors argue that the method is communication efficient since the achieved targeted performance occurs in an earlier communication round for the proposed method compared to other methods. However, I wonder if this is actually the case for the actually communicated number of parameters (for example in Figure7)?
>
> In terms of total communicated parameters, our method is also communication efficient. The table below details the optimal performance alongside the total number of communicated parameters. Our method achieves the best performance with the fewest total communicated parameters.
>
> |            | ROC-AUC | Total # of comm. params |
> |:----------:|:-----:|:-----------------------:|
> |   FedAvg   | 57.18 |          15.66M         |
> |   FedProx  | 57.21 |          15.66M         |
> |    DRFA    | 57.20 |          10.44M         |
> |    FedSR   | 57.25 |          20.88M         |
> | TFL (Ours) | 58.41 |          10.44M         |

---

> > ### Author Response · Authors · 2023-11-22
> > **We are keen to discuss further with you**
> >
> > Dear Reviewer **Nx6V**,
> >
> > We thank you again for your time and your constructive comments.
> >
> > We would really appreciate it if you could kindly let us know whether there are any further questions. We will be more than happy to address them.
> >
> > Best wishes,
> >
> > Authors

---

### Author Response · Authors · 2023-11-23
**General response**

Dear Reviewers, Area Chairs, and Program Chairs,

Thank you for your time and effort in reviewing our paper. We appreciate the reviewers find our studied problem ``”relevant in FL”`` (Nx6V), the proposed framework ``”novel”`` (L1rQ), the experiments are ``”extensive on datasets and baselines”`` (Nx6V, Q4J6, L1rQ, RF1L), ``”cover many aspects of the proposed framework”`` (RF1L), our results verify the ``”superior performance”`` (Q4J6, L1rQ) of our method, the curated datasets are ``”beneficial in future work”`` (RF1L), and the writing is  ``”well-written and easy to follow”`` (Q4J6, L1rQ). We have responded to the individual comments of each reviewer and carefully revised the paper. All revisions are highlighted in light blue.

**Contribution of this work.** **At the problem level**, we investigated the generalization problem for out-of-federation (OOF) scenarios in FL, verifying that existing methods suffer from poor OOF robustness. **At the technical level**, we propose a Topology-aware Federated Learning (TFL) framework, which utilizes client relationships to develop robust models against OOF data. We design an iterative client topology learning and learning on client topology approach to solve TFL. **At the experiment level**, through extensive experiments on standard benchmarks and real-world datasets, we verify TFL’s superior OOF-resiliency and scalability.  Additionally, we contribute to the field by curating two OOF benchmarks using real-world healthcare data, offering valuable testbeds for future research.

**Summary of Reviews:** Below is an overview of the comments received from the reviewers, along with our corresponding responses:
1. For Reviewer **Nx6V**, the primary concern raised was the computational overhead in extremely large-scale settings. In our paper, we address this by discussing the potential of employing a clustering-based method. To assess its effectiveness, we conducted experiments using the eICU dataset, where our results demonstrated that the clustering-based method can significantly lower computational costs by 69%. Regarding concerns about equation (5), we have included a detailed clarification to eliminate any misunderstandings. Lastly, we have included the total number of communicated parameters to illustrate the communication efficiency of our method.
2. Reviewer **Q4J6** recommended providing more detailed explanations regarding influential clients. In response, we have revised the relevant sections of our paper to offer a more comprehensive understanding. Furthermore, we have more discussion on communication efficiency, particularly focusing on the total number of communicated parameters. The reviewer positively acknowledged our response ``“I appreciate the additional table regarding the total number of communication parameters.”``
3. Reviewer **L1rQ** expressed the need to have results of the clustering-based method. In response, we carried out experiments on the eICU dataset and have reported the results. Our results show that the clustering-based method is effective in significantly reducing computation costs by 69%. Moreover, we have detailed discussions on addressing the open question of comparing our method with the prototype-based approach (e.g. FedProto, Tan et al., AAAI 2022), as well as its potential efficacy in managing adversarial clients.
4. Reviewer **RF1L**'s primary concern was centered on the rigor of the notations and theoretical analysis in our work. In response, we have refined our notations for enhanced clarity and consistency. Additionally, following the reviewer's suggestion, we revised the introduction of our paper to broaden its scope, ensuring it doesn't solely focus on specific applications.  Our responses address the reviewer's feedback, as evidenced by the positive comment: ``“The provided response aligns with my expectations, prompting me to raise my score.” ``

We believe that the clarifications and additional results we've incorporated have enhanced the quality of our paper. We respectfully request the reviewers to consider these improvements when re-evaluating and adjusting their scores. We are open to and welcome any further discussions with the reviewers to address any remaining concerns or queries.